# Early Intervention with Parents of Children with Autism Spectrum Disorders: A Review of Programs

**DOI:** 10.3390/children7120294

**Published:** 2020-12-15

**Authors:** Liliana Paulina Rojas-Torres, Yurena Alonso-Esteban, Francisco Alcantud-Marín

**Affiliations:** Department of Developmental and Educational Psychology, Universitat de Valencia, 46010 Valencia, Spain; liliana_ro_6@hotmail.com (L.P.R.-T.); Yurena.Alonso@uv.es (Y.A.-E.)

**Keywords:** autism spectrum disorders, early intervention, parent-mediated intervention, parental training

## Abstract

The aim of this article was to analyze the evidence regarding the effectiveness of intervention programs for children with autism based on the participation of their parents. To obtain the data, a systematic search was carried out in four databases (PsycARTICLES (ProQuest), ERIC (ProQuest), PubMed (ProQuest), and Scopus). The retrieved documents were refined under the inclusion/exclusion criteria, and a total of 51 empirical studies were selected. These studies were first classified according to the function of the intervention objective and, later, by the methodology applied (19 studies were based on comprehensive interventions, 11 focused on the nuclear symptoms of autism spectrum disorder (ASD), 12 focused on the promotion of positive parenting, and nine interactions focused on child play). Once all of the documents had been analyzed, the evidence indicated scientific efficacy in most studies, mainly in those based on child development and the application of behavioral analysis principles. Moreover, the positive influence of parent participation in such programs was demonstrated.

## 1. Introduction

Parents of children with an autism spectrum disorder (ASD) often report alterations in their psychological well-being [1,2,3] and high levels of stress [4,5]. For this reason, in the early attention community centers in Spain, attention has been given to families, in addition to the children, for some time now [6,7,8]. In such settings, family care is understood as guidance, coordination, and accompaniment [8], and providing quality, scientifically proven information is one of the best possible types of support [9,10]. Collaboration with the family and coordination between the professionals that work with ASD have a common objective: an improvement in the quality of life for each family member and for the family system as a whole [11].

The involvement or mediation of parents in the intervention of their children is another step for which many parents are not prepared. The goal of parental involvement in interventions is based on the principle that neuropsychological development is determined by interaction with the environment [12]. Additionally, a child with ASD is characterized by a deficit in basic communication and social interaction skills, and, therefore, may generate inappropriate interaction patterns with his or her parents [13,14,15], thus, resulting in negative effects on their neuropsychological development, thereby producing a cascade effect [16,17,18]. These inappropriate interactions generate a double loop that, on the one hand, continues to fuel the child’s development by increasingly moving him/her away from normative development, and on the other hand, generates a high level of tension and discomfort in the parents that, in turn, increases inappropriate interactions [19]. The need for training of and attention on parents is made necessary, among other reasons, by the evidence found pertaining to parental stress related to their participation in their child’s treatment process [20].

An early intervention that is properly structured for both the parents and the children, would allow to diminish the cascade effects, resulting in the development of an affected child following an evolution closer to neurotypical development [15,21], since development benefits from greater brain plasticity at this age [22]. Evidence of the relationship between parental behaviors and the development of children with ASD is clear [23]; thus, when interventions are implemented with children who show warning signs of autism and their families, the symptoms are improved. This improvement manifests itself even years later [24]; for example, Kim, Bal, and Lord [25] showed that parental involvement in an early intervention is a good predictor of later academic performance.

The involvement of parents in the implementation of intervention strategies designed to help their children with ASD has a long history [26,27], having been given different names (e.g., parental training or parental education). These terms are more or less vague and include everything from parent–therapist coordination, psycho-education sessions about the disorder, and training on specific techniques for language development or for improving social skills, through to specific programs to address maladaptive behaviors [28] and parent-mediated treatment [29].

Due to the large volume of publications around early interventions in children with ASD and their families, it is necessary to carry out periodic systematic reviews to organize the different results. Among the precursors to this study are those of Diggle and McConachie [26,30,31].

In the present work, we conducted a systematic analysis of the scientific documentation about parents’ participation in early care programs. This study was carried out based on the principles of a systematic review, where the criteria of the search, selection, and evaluation of the documents of PRISMA were considered [32].

## 2. Method

Four databases were searched (PsycARTICLES (ProQuestProQuest, LLC, 789 E. Eisenhower Parkway, Ann Arbor, Míchigan 48106-1346 USA), ERIC (ProQuest), PubMed (ProQuest), and Scopus). Only those published in scientific journals subjected to double-blind peer review were selected. The search was performed by accessing all of the databases using the online search interface TROBES of the Documentation and Library Service of the University of Valencia (Spain).

### 2.1. Search Strategy

The search terms used were “autism” OR “pervasive developmental disorders” AND “early intervention” AND “parent training” OR “parental Teaching.” These terms could appear anywhere in the indexed document. The search ended in December 2019.

### 2.2. Inclusion Criteria

The selection was delimited to the period between the years 2010 and 2019. Only those articles that offered empirical data on the results of an intervention were included. Of the selected articles, the bibliography was analyzed using the so-called snowball technique in order to detect any other non-indexed study that could provide some relevant information. In order to keep the information as updated as possible, a search alert in Google Scholar was activated during the time of document review and writing of the paper. The inclusion of documents was completed in September 2020.

### 2.3. Exclusion Criteria

A total of 1010 articles were found following the search criteria, of which, 98 duplicate articles were found and a further 424 articles were excluded because they corresponded to guides and other documents. During the screening process, the abstracts of the articles were read and those that met the exclusion criteria (out-of-age zero to six years, non-empirical studies, studies in which parents did not participate, directed at other disorders), were excluded another 227 documents that dealt with studies on parenting in other areas. Later, during the eligibility process, the entire text of the papers selected in the previous step was reviewed, applying the same exclusion criteria. Finally, 51 articles remained, in which empirical studies were presented with intervention models involving parental participation. Figure 1 shows the flow of the search process.

### 2.4. Quality Assessment

For the evaluation of the evidence, we chose to follow the initiative of the *Journal of Clinical Child and Adolescent Psychology* (JCCAP), in particular, the proposed adaptation [33] to evaluate the evidence of treatment in children with ASD.

The criteria we propose for this work (see Table 1 and Table 2) distinguish between studies with well-established or adequate evidence (Level 1) and studies with probable or possibly effective results (Level 2). Among the former, at the same time, the criteria differentiate between methods that, due to the number of published clinical trials (randomized controlled trials (RCT)), can be analyzed as joint statistical results or as meta-analyses and those methods that only have an RCT, albeit a very robust trial because it has a wide stratified sample. Meanwhile, among Level 2 studies, the criteria also distinguish between RCT studies with small samples and meta-analyses of series of single case studies.

Evidence from quasi-experimental studies (Level 3) are those that have not yet reached adequate levels of evidence, but the results can point to new studies in the future in a positive direction. These are generally quasi-experimental studies in which one or two groups are measured. Their fundamental characteristics are simplicity and economy in development. They can be grouped into two types, namely, single groups with pre- and post-tests and several groups; the latter differ from the experimental groups in that the subjects are part of natural non-random groups. These types of studies are not conclusive, but they can be a powerful tool, especially when randomized experiments are not yet possible. They allow an overview and follow-up to determine or confirm the reasons for the results found [34]. Case studies can also be included at this level. Single case study (SCS) research is experimental and aims to document the relationships between the independent variable (experimental treatment) and the dependent variables. Since it is a single case, the individual differences that affect the internal validity of the experience are controlled. The accumulation of the results of different SCSs on the same problem and with the same method can increase the evidence [35], thereby increasing the external validity. Finally, there are studies (Level 4) that could be described as providing questionable evidence due to the characteristics of the method used.

## 3. Results

In total, 51 studies offering empirical results about the role of parents in 15 intervention programs with different approaches were included, which we classified into four large groups: (a) participation in comprehensive programs, (b) participation in programs targeting the core symptoms of ASD, (c) participation in programs aimed at improving parent–child interaction, and (d) participation in parent–child play-based programs (see Table 3).

### 3.1. Parent-Mediated Intervention in Comprehensive Intervention Programs

Comprehensive intervention programs are those that address all of the core symptoms of ASD; therefore, they aim to develop social skills and interests, address communication difficulties, and reduce repetitive, ritualistic, or stereotypical behaviors. They are distinguished from other intervention programs that specifically address communication deficits (e.g., Picture Exchange Communication System (PECS) [86], aberrant behaviors [87], self-injurious behaviors [88], or eating disorders [89]). In our study, we found documents related to parental involvement in four of these programs.

#### 3.1.1. Parental Training (PT)

Training programs in Applied Behavior Analysis (ABA) principles are called parent-training programs (PTs) [90]. The initial objective was directed at the extinction of disruptive behavior [91], but soon after, programs were also developed for the development of social, communication, initiation, and language skills [92], with the aim of reducing behavioral problems.

A total of eight studies were found that provided medium or medium–high evidence of the efficacy of this intervention (see Table 3). Oosterling et al. [36] developed a clinical trial to compare the results after 12 months of training with parents in two groups, one complementary to the usual intervention and the other without intervention. The training focused on joint attention and language development. In total, the group consisted of 75 children (28–42 months old). No significant differences were found, so it was concluded in this study that parent training does not add value to the overall intervention.

Bearss et al. [37] proposed another trial with 16 children (three to six years old) presenting with ASD with disruptive behaviors. The intervention was prolonged for six months and was very well accepted by the parents (84% of them finished the program). Among the results, a decrease in the scores of aberrant behaviors and irritability stands out. In a later replication [28], a randomized test was conducted to measure the effectiveness of a program mediated by the parents of children with ASD with behavioral problems. The study lasted 24 weeks and involved 180 children (aged between three and seven years) and their families. The results showed that a parenting program, such as PTs, can help reduce disruptive behaviors. 

Video modeling has been shown to be a cost-effective and efficient tool in many cases. Bagaiolo et al. [38] presented a clinical trial with a control group consisting of 67 parents of children with an ASD diagnosis aged three to six years who attended PTs to improve their social behavior (i.e., disruptive behaviors) as part of an ABA intervention. Twenty-two working sessions were designed, in which video modeling was used in one group but not in the control group. They concluded that the video modeling method did not introduce negative effects, but rather resulted in positive ones, showing that it is a possible and low-cost form of intervention, particularly in populations with scarce economic resources. In contrast, the impact from the use of information and communication technology (ICT) in recent years is currently being studied, particularly the replacement of live consultation sessions with synchronous video calls to avoid geographical problems arising from the dissemination of the rural population [39]. Blackman et al. [40] presented a clinical trial formed by three groups of parents who received PTs (i.e., online, in vivo, and a waiting list control group) in ABA. The results showed that both training methods were effective and suggested that online asynchronous training can serve as a cost-effective alternative in ABA PTs. The content included video recordings, readings, and training modules for parents created by the professionals. Iadarola et al. [41] presented a comparative study between the results of a PT program and a psychoeducation program. In total, the results of 180 children and their families were compared, and different measures of parental stress, effort, and caregiver competence were evaluated. This is possibly the largest PT program trial to date, and it showed that PTs reduce children’s disruptive behaviors, improve their competencies, and decrease parental stress and tension.

#### 3.1.2. Pivotal Response Training (PRT)

PRT is a program derived from and developed under the paradigms of ABA methodology, and, to some extent, is an evolution of it, aiming to help solve the problems of generalization [93]. It focuses on fundamental areas or skills (pivotal areas) under the hypothesis that an improvement in these areas will produce improvements not only in the areas worked on, but also in other functioning areas [94]. Parents or caregivers play an active role in the treatment by helping to carry out the intervention [94]. In PRT programs, parents should attend training programs in which they learn techniques and ways to improve their child’s motivation and self-initiation through communication and academic skills [94]. In the period analyzed herein, four studies were found.

Minjarez et al. [42] developed a trial with the aim of evidencing the possibility of applying parent-mediated PRT. They selected 26 families with children diagnosed with autism aged between two and six years. The treatment consisted of a 10-week training package (90 min group sessions, plus 50 min of personal attention). The program was developed over 18 months and the groups comprised 8–10 parents. As a result, it was noted first that PRT training for parents in groups is beneficial, as it was possible to increase communication between parents and children by improving their language. Consequently, it was considered that group parent training can be incorporated into PRT programs not only efficiently and cost-effectively, but also to improve the generalization of the behaviors learned during the clinical sessions.

Gengoux et al. [43] attempted to answer the question of maintaining parental behaviors beyond the end of the program. They developed a trial with 23 families, with follow-ups over 12 weeks. The empirical results supported the benefits of parental involvement in the implementation of PRT, leading to improvements in their children’s language and cognitive function, and these benefits were maintained for at least 12 weeks after treatment.

In the same year, Hardan et al. [44] developed a new clinical trial with 53 families with children with ASD aged two to six years. A 12-week group-training program in PRT (GTP-PRT) and a psychoeducation program were developed. The results suggested that both parents and children who attended the GTP-PRT training developed more communication skills and adaptive behaviors. Bradshaw [45] presented a single case study looking at the outcomes of three children (17–21 months). A one-hour parent intervention was developed over 12 consecutive weeks in the family home. The intervention focused on the development of expressive verbal communication. The results showed an important increase in the number of words used by the children in their communication and an increase in the communicative initiations, while parents reported high levels of satisfaction with the program.

#### 3.1.3. Treatment and Education of Autistic and Related Communication-Handicapped Children (TEACCH)

TEACCH [95,96] is a philosophy developed in the state of North Carolina (USA), approved by state parliament as a guide for the lifelong care of people with ASD. Schopler et al. [97,98] defined the role of parents as necessary partners, creating a relationship between parents and professionals that is essential and central to treatment. Specific training actions were developed based on the characteristics of the disorder, instructed on measures to reduce children’s difficulties (i.e., continuous and structured intervention, an adaptation of environments, and use of alternative and augmentative communication systems). The technique that has transcended the most and for which TEACCH is recognized is structured learning [96]. The premise behind this intervention is to modify the context to meet the needs of the individual with ASD. To do this, it adapts the environment, collaborating with parents, evaluating treatment outcomes, and providing generalist training. The results of TEACCH intervention programs developed by parents in the family home have been positively evaluated [99]. However, randomized clinical studies on TEACCH are not very abundant [100,101], as seen in Table 3.

Welterlin et al. [102] developed a study to evaluate the effectiveness of a TEACCH-based intervention program conducted in the family home. A total of 20 families were assigned to the intervention groups or to the waiting list. The results showed an improvement in both the children’s behavior and the parents’ skills. However, due to the small sample size, the study was not conclusive.

D’Elia et al. [46] conducted a follow-up study for the application of TEACCH in schools and educational centers, evaluating the level of severity of the disorder, adaptive functioning, language, aberrant behaviors, and parental stress. The results suggested that a combined home and school intervention provides benefits to children with ASD by reducing the intensity of their symptoms and aberrant or maladaptive behaviors. In addition, reducing parental stress becomes a crucial factor in the success and effectiveness of the program.

#### 3.1.4. Early Start Denver Model (ESDM)

ESDM (https://autismcenter.duke.edu/ and https://www.esdm.co/) [47] is a methodology based on behavioral principles that takes into account the typical development of a child and was created due to the need for early and intensive interventions. This program was designed for children between 12 and 60 months of age. In it, parents receive training from professionals to replicate what they have learned in consultation with other contexts. The main objective is to achieve progress in functional development, social skills, and language development, as well as to increase attention and motivation and to improve family dynamics. The results of the search (see Table 3) showed a low number of studies pertaining to this type of intervention, almost all of which were carried out by the same research team.

Rogers et al. [103] developed a study to examine the effectiveness of the application of a parent-oriented ESDM program (P-ESDM). In this program, the emphasis was placed on promoting receptive, child-centered interaction styles and incorporating more play opportunities than conventional treatments. This research involved 98 children at risk of ASD between the ages of 12 and 24 months and their caregivers. For the application, participants were divided into two groups at random, namely, an experimental group and a control group. The results showed no significant differences in the children who participated in the experimental group; meanwhile, the parents of the two groups showed improvements in their interaction skills, although the parents who received the EDSM intervention achieved greater adherence to treatment. It should be noted that the control group (conventional treatment) received more hours of treatment with a therapist than the experimental group, so we can conclude that the program with parents was effective.

Rogers et al. [48] followed this line of research by developing a study during the first year of life of seven children. The main objective of this study was to develop and test the reliability of a parent-guided intervention. The intervention targeted the reduction or modification of six symptoms and developmental patterns of autism in the early stages. For this study, the participants were divided into two groups, namely, an experimental group and a control group; four of the children were assigned to the experimental group (diagnosed ASD) and the remaining three (control group) were at risk of autism only. The results showed that during the first nine months of the intervention, the two groups had the same changes, but when they were in the 18–36-month period, the experimental group achieved greater changes in symptomatology, language, and visual response. Likewise, it was observed that the parents were able to acquire skills for the management of their children, thereby improving the parent–child relationships.

Vivanti et al. [49] analyzed the effectiveness of the ESDM program applied to children diagnosed with ASD between the ages of 18 and 60 months. Participants were divided into two groups: 27 formed the experimental group that received the ESDM program, while the other 30 formed the control group. The trial lasted 12 weeks with 15–25 h/week of intervention. The parents, however, received training in six two-hour sessions on the ESDM strategies with the aim of being implemented at home in daily tasks. The results obtained were positive and demonstrated not only that ESDM is suitable for the treatment of ASD, but also that this study achieved more reliable and stronger data than previous research. Unfortunately, no action was taken, and no record was made of the parents’ intervention at home.

Based on the development, two lines of work were created: The conventional ESDM model and the P-ESDM model. Rogers et al. [50] carried out a randomized comparative study to see if the P-ESDM version could generate greater changes in participants. For this research, 45 children diagnosed with ASD who were between the ages of 12 and 30 months were selected. All of the children were assigned to one of two groups in a randomized fashion receiving the same intervention (12 sessions of 1.5 h/week). The P-ESDM group had the same hourly intensity, adding one and a half hours per week of work at home. From the results of this study, it was demonstrated that ESDM is effective, since significant improvements in the functional development of children were observed in both groups. In addition, the P-ESDM group had better results in terms of parent–child interaction skills. This improvement was associated with greater individual progress of the children regarding qualitative measures, although not on standardized assessments. 

The last study found [51] was on five mothers of children with ASD under the age of five, and the intervention was based on a 12-week ESDM parent-training program. The goal of the intervention was to conduct parent training within the home where the children were to be present. Direct instruction, modeling, skill practice, and feedback were used to achieve this training. Although there was great variability in the results of each mother/child dyad, the results showed that the mothers learned to use the techniques of the model, generating positive changes in their children; among these changes were the management of unwanted behaviors, greater commitment in the children, and improvements in expressive language.

### 3.2. Parent-Mediated Programs Focused on ASD Symptoms

The influence of the family environment on the development of socialization processes and communication and language development has long been known [52]. This influence is reflected in the relationship between the styles and quality of parent–child interactions and cognitive development, language, and social skills in both typically developing children [104] and those at risk of ASD [105]. There are several programs for improving the relationship, dynamics, and communication between parents and children with therapeutic goals. In our search, we detected a total of four such programs.

#### 3.2.1. Hanen More Than Words

The Hanen program (http://www.hanen.org/Home.aspx) [106] with more than 35 years of experience is perhaps one of the oldest. Hanen’s “More Than Words” and “It Takes Two to Talk” are two programs designed to improve family dynamics and parent–child communication. These are general purpose programs not specific to children with ASD, and they can be used to improve the family dynamics in families with children with language delays [107,108], intellectual disabilities, motor disorders [109], or ASD, or in families with internal or dysfunctional relationship problems. The number of studies found (see Table 3) shows that even though it is a very experienced program, it lacked evidence during the studied period.

Carter et al. [110] developed a randomized trial comparing the Hanen program with conventional treatment. A total of 62 children diagnosed with ASD participated, and their language and communication levels and the parental responsibility of the parents were assessed. The effects of the program for parents showed differential effects according to the initial profiles of the children.

#### 3.2.2. Preschool Autism Communication Trial (PACT)

PACT [53] is an intervention program that aims to improve communication between parents and children with autism, directly affecting the social and language development of said children, which was developed at the University of Manchester, United Kingdom. The first trial was conducted between 2006 and 2009. The aim is to train parents to adapt their communication style to their child’s abilities and to respond to their child with greater sensitivity and responsiveness. The emphasis in the program is on increasing joint attention through looking or sharing, showing, and giving, adapting the language to the child’s level. Different strategies are also presented to facilitate communication and child participation (routines, verbal scripts, use of elaborations, pauses, etc.). Through this training and with different adaptations, parental sensitivity, and positive interactions within the family context are increased. In the period analyzed, only one clinical trial related to PACT was found (see Table 3).

Green et al. [53] conducted a RCT with 152 children between two and five years old. The children and their families received the usual treatment in three specialized centers in the United Kingdom. Parents of the PACT group received more training consisting of an initial meeting and two-hour clinical sessions over six months. At 13 months, the severity of the symptoms of the children in the PACT group was reduced by three to nine points assessed by the ADOS-G (Autism Diagnostic Observation Schedule Generic) algorithm, while in the group assigned to conventional treatment the improvement was less. In conclusion, although the application of PACT is not systematically recommended to conventional treatment, the help provided by such an intervention is recognized, especially in social and communication areas. Given these results, Pickles et al. [54] reanalyzed the results of Green et al. [53], proposing a mediation model to understand the relationships between parent and child behaviors. In a subsequent follow-up study on the same sample [111], they observed that there was an improvement in the dyadic social communication between parents and children, although no relationship with the aims of conventional intervention on the nuclear symptoms of the disorder was observed. Nevertheless, the improvement in communication attenuated behavioral problems in the family that were retained in the long-term. 

#### 3.2.3. Joint Attention Symbolic Play, Engagement, and Regulation (JASPER)

JASPER [112,113], developed at the “Center for Autism Research and Treatment, University of California Los Angeles” (https://www.semel.ucla.edu/autism), builds on previous research team studies where deficits in joint attention and symbolic play were found to be two of the most important developmental issues for children with ASD [112,114]. Kasari et al. [113] developed clinical trials with parents and caregivers as mediating agents and follow-up studies [115], in which the relationship between joint attention, symbolic play, and later language development was evidenced by developing the JASPER program. This is an intervention program that focuses on the fundamentals of social communication and uses naturalistic strategies to increase the pace and complexity of the social relationship. Its objective is to increase social commitment, verbal and non-verbal communication, and skills during play based on parental education, which generates commitment from parents to strengthen these areas through motivating and enjoyable activities. As shown in Table 3, the number of studies on this subject is limited and, in most cases, they were developed by the same research team.

Goods et al. [116] developed a clinical trial on minimally verbal children where they evaluated the incorporation of JASPER sessions over a conventional ABA program. The intervention was developed across 12 weeks, in which the control group only received conventional sessions of the ABA program, while in the experimental group, 30 min were substituted with JASPER sessions, demonstrating that those attending the experimental group experienced greater increases in play and initiation of communicative gestures.

Providing continuity to their research, Kasari et al. [55] carried out a comparative study between the JASPER model and a psychoeducational intervention for parents. The participants were 86 children in an age range of 22 to 36 months and their primary caregiver. For this study, the dyads were divided into two groups randomly. The aim of this research was to determine if the JASPER methodology has greater results in stress management and behavior control in children. The results showed that the JASPER group obtained significant and greater effects than the children in the control group. Among the gains that were observed were high-level relationships during play, joint attention, engagement, and social initiation.

Following the study by Kasari et al. [55], Gurlsrud et al. [56] carried out a second clinical trial to determine the influence of JASPER’s components on increasing behavior management skills and strategies and whether this influences social engagement. To test their hypothesis, they applied the intervention to 86 children under 36 months old, who were divided into two groups, namely, an experimental group and a control group. The results obtained were positive, since it was possible to determine the four central strategies of the intervention and the role of the parents, demonstrating that this type of intervention positively influences parent–child relationships and that there is a significant increase in joint commitment.

In the last study found, Shiere, Gulsrud, and Kasari [57] compared the application of JASPER and parent education intervention programs in order to determine which of the two intervention models generates greater changes in behavior, social communication, and commitment of both parents and children. To carry out this research, 85 children (under 36 months old) and their caregivers participated. The results showed no clinically significant differences between the two groups, as all children showed gains in language use and social engagement. However, it was evident that the group of parents belonging to the JASPER intervention had changes in behavior that directly influenced the relationships with their children.

#### 3.2.4. Improving Parents as Communication Teachers (ImPact)

ImPact [58,117] is a program designed to integrate parents and teachers in the early intervention of children with ASD, developed in the “Autism Research Lab” at Michigan State University (http://psychology.psy.msu.edu/autismlab/projectimpact.html), USA. It is based on numerous previous studies of the research group in which social communication [118], imitation [119,120,121,122] in the social development of children with ASD, and parental involvement in the intervention [117,123] were important. Combining these elements through parent training in communication skills with their children promotes the generalization of children’s skills, increases parent optimism, and decreases stress [123]. Based on naturalistic and developmental behavioral intervention strategies [124], a program manual for parents and educators was developed to promote child social engagement, language, imitation, and play during daily routines and activities. In Table 3, it is shown how the studies found on this topic were mostly conducted by the same research team.

In our review, we found four empirical studies, the first of which was by Ingersoll and Wainer [125], who created a trial to evaluate the effectiveness of the ImPact program on children attending public special education centers. In total, the intervention was initiated with 30 teachers who invited the family to take part in the program. Ultimately, only 24 families completed the program. Among the results, we highlighted a decrease in parental stress and an increase in social communicative response with an increase in the use of language.

Following on in the search for evidence, Ingersoll and Wainer [59] published a new study based on a single case design in which they accumulated a total of eight preschool children with ASD. As parents increased their use of intervention techniques, an increase in spontaneous language use was observed in six of the eight children. This suggests that there is a relationship between the use of intervention strategies and language use in children.

Another team [60] developed an ImPact trial under a community program, conducted over 12 weeks and applied to 30 children (two and a half to six and a half years old) and their parents (two groups: 16 in the intervention group and 14 in the control group). The study showed improvements in the children’s social communication styles, as well as an increase in the same direction of parental adherence to treatment and a decrease in stress. These results suggest that ImPact adds positive effects to conventional community interventions, and is therefore recommended.

In order to test if tele-assistance could be a suitable tool to overcome the obstacles of geographical location and distance to treatment center, Ingersol et al. [61] conducted a trial comparing the results in two ImPact parent-mediated treatment groups. In the first, web-assisted self-implementation strategies were applied for six months. The password-protected URL contained 12 self-administration lessons (approximately 75 min each). The second therapist-assisted group had the same structure and duration of web access, but received two additional 30 min sessions per week of support from an expert therapist via video conference. Both groups improved their results (parents and children), although these results were better in the group that received support via video conferences. In addition, 100% of that group completed treatment, while only 65% of the self-administered group did.

The last of the studies [62] attempted to determine the value of low-intensity intervention (1.2 h/week) without including parent training. A single case study was presented with a cumulative total of nine children with ASD (three to eight years old). Although wide variations were observed among the children, all of them showed improvements in two or more intervention areas (expressive vocabulary, social engagement, etc.).

### 3.3. Programs for the Promotion of Positive Parenthood and Family Well-Being

Related to the previous section, if neuropsychological development is determined by interaction with the environment [12] and a child with ASD has a deficit in basic communication and social interaction skills with his or her parents, it may generate inadequate parental interaction patterns, among other reasons due to the stress generated by the new situation [13]. Families (parents) who are faced with raising a child with a developmental disorder suffer a greater number of sources of stress and, in turn, this manifests itself with much more intensity [63,126]. Parents of children with ASD often report changes in psychological well-being [1,2,3] and high levels of stress [4,5] by altering patterns of parent–child interaction [14,15]. In our search, we found four programs that aim to improve parenting and overall psychological well-being, as well as children’s symptoms.

#### 3.3.1. Parent–Child Interaction Therapy (PCIT)

PCIT [127] is defined as a brief therapy program based on behavioral principles, and is directed at solving behavioral problems of parent–child interaction. It is perhaps one of the most recognized behavioral training programs for parents. Originally, PCIT was used to solve the problems of disruptive behavior and disobedience in children, but it has also shown good results in language development and emotional recognition [128]. It has also been applied in families with children with ASD [129,130]. Within the study period, we found five PCIT-related studies developed by different teams (see Table 3).

To demonstrate the effectiveness of this model in children with ASD, Lesack, Bearss, and Celano [131] implemented a PCIT-based intervention (with adaptations) with a five-year-old boy diagnosed with autism who presented difficulties in expressive and receptive communication and behavioral problems. The results showed clinically significant reductions in disruptive behaviors, gains in the child’s functional development, and increases in parenting skills. The mother reported increased use of commands and communication by her child, greater engagement, and a better quality mother–child relationship.

The influence of PCIT on the development of vocalizations has also been studied. Hansen and Shillingsburg [64] proposed a study under the PCIT model to determine its influence on increasing vocalizations. For this purpose, a single case study was carried out with two children diagnosed with ASD aged 45 and 32 months. The results in both cases showed that the children increased their total number of vocalizations and the parents reported high levels of satisfaction and acceptability of the program, as well as improvements in their children’s language and functional behavior.

Another single case study [65] was conducted to examine the effectiveness of this program. The authors found that children had reductions in disruptive behaviors, increased parent–child communication, and, in two of the three cases, increased compliance with parental demands. Parents also expressed high satisfaction with the program, suggesting that this methodology may be a treatment option for children with ASD who present behavioral difficulties.

A new study of 17 children with behavioral and diagnostic problems has since been published [66]. The goal was to apply the PCIT model in order to determine the effectiveness and reliability of the program and to examine changes in behavior. The results showed significant reductions in disruptive behaviors and the strengthening of parenting skills. In addition, parents reported that their children had increased levels of functional development, communication, and pro-social behavior.

Finally, Parladé et al. [67] performed a study to examine the influence of a PCIT-based intervention on children with behavioral problems and ASD. The goal was to observe changes that may occur in parenting skills, parental stress, and child behaviors. For this purpose, 36 families with children aged three to seven years were recruited, who were divided into two groups: The experimental group included children with a confirmed diagnosis of ASD, while the control group included 20 children with behavioral problems. The results showed that this program helps to reduce the occurrence of behavioral problems in typically developing children with ASD. It was also shown that children with autism were able to decrease autistic symptoms and obtain improvements in social response, social skills, adaptability, and repetitive and restrictive behaviors.

#### 3.3.2. Prevent–Teach–Reinforce (PTR)

PTR [68] is a model of positive behavior support (PBS) designed to be applied in school environments with the support of family members [132,133], which has also been successfully tested in families (PRT-F) [134], with children with developmental disorders [69], and particularly with families with children diagnosed with ASD [134].

In the study conducted by Sears et al. [134], the PRT program was administered to two children with ASD aged four and six years and their caregivers; the main goal was to examine the effectiveness of the implementation of the PRT program on children diagnosed with ASD. The program was implemented in each child’s home and the intervention was led by the parents who had received training on the skills to use. The results showed that the PTR program can be adapted and implemented at home and conducted by caregivers, and there was evidence that both families successfully created and implemented behavioral plans. Moreover, there was a reduction in disruptive behaviors and the appearance of proper behaviors in the children during the intervention.

Meanwhile, Bailey and Blair [69] analyzed the limitations of the PRT model at the time of collecting data to prove its validity. They make a replica of Sears et al.’s [134] study, ensuring the collection of data using the Individualized Behavior Rating Scale Tool (IBRST). Three families of children with ASD and language delay with sensory problems (five to seven years) were invited to participate. The results showed that both families and children achieved high levels of adherence to the program and learned to apply the intervention successfully within the home. During the intervention, a dramatic decrease in negative behaviors was observed, and with it the emergence of appropriate behaviors.

#### 3.3.3. Collaborative Model for Promoting Competence and Success (COMPASS)

COMPASS [70,135] is a program designed as a conceptual framework for planning responses to individual needs identified by teachers for students with ASD. Trials have also been developed comparing face-to-face with web-based forms of the intervention [136]. To increase the effectiveness of the program, parents have also been included [137]. The goal of parent training and family support programs is to increase family competence and to establish positive parent–child interactions, thereby achieving a decrease in the occurrence of parental stress [71].

In the study by Ruble et al. [137], the COMPASS program was implemented in collaboration with the teachers and parents of children with autism spectrum disorder. The sample used comprised 35 parents, teachers, and children, which were divided into an experimental group (including teachers trained in COMPASS) and a control group. The results were not very strong because no clinically significant differences were found between the two groups. However, it could be concluded that collaboration with teachers can help children with autism be part of educational environments more adapted to their needs.

Joining the experience of telematics assistance for teachers [136] and parents [137], this same team developed another version COMPASS for Hope (C-HOPE), whose objective was the reduction of parental stress [71]. To demonstrate the changes, they conducted a randomized clinical trial on 33 families. A significant reduction in parental stress and an increase in parental competence were detected. Parents also reported a significant reduction in their children’s behavioral problems, both when comparing the rates with previous levels and when comparing them with the waiting list control group. The treatment modality (online or face-to-face) did not produce significant differences.

#### 3.3.4. Stepping Stones Triple P (SSTP)

SSTP [72] is a parenting program designed for families of children with a disability based on the standard Triple P (TP; Positive Parenting Program) [138,139], which was developed by the Parenting and Family Support Centre, The University of Queensland (https://pfsc.psychology.uq.edu.au/), Australia. Stepping Stones, a variation of the parenting training program, shares strategies focused on the processes of acquiring concrete skills such as communication using ABA principles, as well as affective development for parents. It is designed specifically for parents of children with disabilities, including ASD. 

Roux, Sofronoff, and Sanders [140] performed a trial based on the group-developed SSTP (GSSTP) methodology with 52 parents and children with ASD, Down’s syndrome, cerebral palsy, and intellectual disability. Participants were divided into two groups (intervention and waiting list). The objective of the study was to demonstrate if such an intervention has positive effects on children’s behavioral problems and if the program achieves improvements in parenting styles. Additionally, the authors wanted to evaluate parents’ perceptions of the program. The results indicated that it is a promising intervention for a mixed disability group, since significant improvements in the children’s behavior and parenting styles and high parental satisfaction with the program were demonstrated.

SSTP is a five-level intervention system with different programs that vary in intensity. Tellegen and Sanders [73] developed a randomized controlled trial to evaluate the efficacy of a short SSTP program (four sessions), applicable in primary care (i.e., Primary Care Stepping Stones Tripe P (PCSSTP)). They selected 74 families with children diagnosed with ASD who were between the ages of two and nine years. The families were divided into two groups (intervention and control). To determine the effectiveness of the intervention, they were evaluated at three stages (pre-intervention, post-intervention, and six-month follow-up). The results showed improvements in the behavior of the children in the intervention group, improvements in the level of parental stress, a decrease in marital conflict, and an increase in general well-being. However, no significant changes were found in the level of depression, anxiety, or parental rejection of the children. The effects were maintained at the six-month follow-up, reporting high levels of satisfaction with the program.

Lastly, Schortt et al. [74] explored the effectiveness of SSTP as complementary to direct intervention for children with ASD. Twenty-two families and children aged 3–12 were recruited to conduct this study. After the intervention, there was a significant reduction in negative parental behaviors, increased parental self-efficacy, and reduced caregiver stress. It was concluded that this type of methodology can be used as a complementary intervention and can be highly effective in the treatment of children with autism.

### 3.4. Play-Focused Intervention Programs

This section refers to methodologies in which games are used as an essential part of the intervention. The value of play in children’s psychological development has been known for a long time [75]. Its application at a therapeutic level has also been recognized for some time [141,142]. Play is a universal activity in all children, through which they rehearse problem situations, so we can consider it key to developing social behaviors [143]. Intervention techniques focused on play with children with ASD, have been used for a long time, and its effectiveness has been demonstrated in meta-analyses [144,145,146], with results that demonstrate changes produced in social–emotional and communication development. Most of the techniques that focus on play are part of interventions based on pragmatic social development (e.g., Developmental Social–Pragmatic (DSP)) model [147]. In our search, we found three programs in which play is the basic tool of the intervention.

#### 3.4.1. Theraplay

Theraplay [148,149] is a play therapy approach designed to improve parents’ attachment, attunement, and sensitivity, as well as children’s regulation and reflection. It focuses on the non-verbal aspects of children’s communication, using playful interactions as a means of intervention. The intervention is carried out in a family context with a duration of 30 min in weekly sessions across a four to six-month period.

A study was found that determined the effectiveness of this methodology [150]. This study was conducted on eight children diagnosed with autism between the ages of three and nine. The intervention was intensive, targeted at the children and their primary caregiver, and consisted of one-hour interventions each day for two weeks. The objectives of the study were organized in three sections, the first of which corresponded to the observation of parent–child interactions; the second determined the changes in the quality of the interactions, and finally, the influence of the intervention on the families’ behaviors was evaluated. The results showed that both parents and their children achieved significant improvements in their interactions and acquired new tools to achieve positive interactions.

#### 3.4.2. Floortime Play

Floortime Play [76] is the practical form of intervention based on the Developmental, Individual Difference, Relationship (DIR©)-based model [76,151,152]. It consists of the development or encouragement of spontaneous and structured or unstructured play sessions, in which relationships are built and self-regulation, two-way communication, social engagement, complex thinking, and problem solving are developed.

Dionne and Mastini [153] presented a unique case study: A three-year-old child and a six-month-old child diagnosed with autism at two years and five months, respectively. Four sessions were conducted across a 7-week period (45 min per session), and all interventions were conducted jointly by the therapists and the mother of the child. Specific observational measures were used to evaluate the intervention. A total of 28 sessions were conducted over the seven weeks of the intervention. The results showed improvements in this variable, as well as in spontaneous communication, family relationships, and exchanges during communication.

For their part, Pajareya and Nopmaneejumruslers [77] conducted a randomized controlled pilot trial to determine the possible additional benefits that this method could bring to routine interventions. Two groups were organized, the first of which only received conventional treatment, while the second group received supplementary DIR©-based Floortime Play sessions. Thirty-two participants were assigned to each group using stratified random assignment according to age and severity of symptoms. The results reflected an overall improvement in ASD symptom severity for all children in both groups, with the improvement being most significant in the DIR©-based Floortime Play intervention group. Similarly, changes were observed in the emotional development of the children participating in the experimental group.

Continuing their research and based on the results of the participants in the intervention group of the previous study, Pajareya and Nopmaneejumruslers [78] conducted a follow-up study to demonstrate the effect of maintenance and adherence to treatment by parents for one year under the home-based care model (i.e., Home-Based DIR©-based Floortime Play). The results pointed in the same direction as the original study; in addition to having achieved improvements in the scores of each of the scales in the pre- and post-test contrast, parents continued to relate positively to their children, which led to improvements in family relationships.

Liao et al. [79] developed a home intervention program based on the DIR©-based Floortime Play principles with the intention of enhancing social interaction and adaptive functioning. The participants comprised 11 children diagnosed with ASD, and the intervention lasted 10 weeks. The program included three weeks of training for the mothers in individual sessions in which individual goals were developed for each child. At the end of the intervention conducted by the mothers, significant gains in communication, life skills, and social skills were achieved.

Solomon et al. [80] developed Play Project Home based on the DIR©-based Floortime Play principles and conduced a controlled trial on a total of 128 families with a child diagnosed with autism. They were randomly assigned to two groups of 64 families, stratifying by age and severity level. The control group continued to receive standard treatment in community services, while the experimental group received Play Project Home training. From the data obtained, the two groups demonstrated improvements in diagnosis, but the group that received the intervention showed greater improvements, and these were statistically significant. The results also determined changes in parent–child interactions, functional development, stress, and depression in parents. Regarding the parent–child interactions, the results indicated that the parents of the experimental program showed a significantly greater change in the quality of their interactions. Moreover, in terms of functional development, the experimental group showed greater changes than the control group, while the results of the parents did not show differences in terms of the levels of stress between the two groups.

Aali et al. [81] presented an experience in Mashhad (Iran) in which they designed a family-centered intervention in combination with DIR©-based Floortime Play. A total of 12 children from two to eight years old and their families participated during the five-month intervention. Although the study referred to the three groups included as independent groups (Family-Centered Therapy group, DIR©-based Floortime Play group, and control group), no detailed information could be found about the composition of these groups or the mechanism of assignment to each of them. The results mentioned gains in the areas of intimacy, commitment, emotional development, and self-regulation; however, the data were inconclusive, and although scales such as the FEAS (Functional-Emotional Assessment Scale) were used, they were limited to making qualitative assessments.

Finally, Sealy and Glovinsky [82] performed a controlled clinical trial in Barbados with parent–child dyads. All of the children presented neurodevelopmental disorders related to communication and relationships, and were between two and seven years of age. A total of 40 dyads participated in the 12-week trial. The objective was to evaluate the developmental changes in reflective parental functioning as assessed by the Parent Development Interview (PDI) [83]. Parent training in DIR©-based Floortime Play improved their reflective functioning skills, suggesting that they learned to better read their children’s social demands and respond accordingly.

#### 3.4.3. Focus Playtime Intervention (FPI)

Siller and Sigman [154,155] published studies that showed that parents’ responsible behavior during their play with their children with ASD in the early years predicts later language development. Based on these studies, they developed an experimental program that they called Focus Playtime Intervention (FPI) [156], which is composed of 12 training sessions in the family home (one per week) of 90 min. Each session is divided into two parts; during the first part, the therapist provides a standard toy pack. The parents and children are invited to take out the toys, and the professional guide is then incorporated into the interaction. The parents and the professional guide alternate in their interactions with the child, demonstrating different strategies. During the second part of the intervention, parents receive instruction on what happened in the session and plan tasks for the week ahead. The results showed that children who start the program at the 12-month language level benefit the most.

In a later reanalysis, Siller et al. [84] determined that parental involvement is relevant because parents must replicate the strategies within the home. Additionally, modeling positive parenting styles allows for improved dynamics within the home, so they understand that PIF intervention should be part of a broader parent-mediated intervention program. The results indicated that not only are parent-child relationships improved, but children’s cognitive development, language use, and independence are also increased.

## 4. Discussion

A total of 51 documents offering empirical data on a total of 15 intervention programs were studied herein. The programs differ in their objectives although all of them have as a common factor the formation of the parents. Four major groups of programs have been described according to their objectives (see Figure 2).

There are also great differences in the level of evidence among the fifteen programs, being especially relevant the absence of a common methodology to evaluate the results. Most programs use different evaluation tools [85]. Evidence supports the effectiveness of techniques and methods based on child development, as well as the application of principles of behavioral analysis [157], including programs that emphasize the use of structured learning environments, stimulus control, development of routines, natural environments, etc. [158]. Evidence also suggests that not all children respond in the same way to all treatments or techniques [159], and that there may be other variables that affect the effectiveness of programs, regardless of the age at the beginning of the treatment [160,161,162], or the intensity [163,164,165]. The role of parents and caregivers as a success agent in early interventions has also been highlighted [79,166].

Comprehensive programs have long established parents on their agendas. Most of these programs incorporate training sessions on the characteristics of ASDs, and as a result of increased awareness, parents’ skills in advocating for their children’s rights are increased, generating a sense of empowerment that, in turn, decreases stress and feelings of isolation [167,168,169]. As for the concept of “parental training,” we believe that it is poorly defined or sometimes misused. We associate it with the more behavioral comprehensive programs.

The Parental Training Program derived from ABA stands out for the number of works and amount of evidence. Moreover, the studies on the P-ESDM parent-mediated model are also very relevant. Among the programs centered on the central symptoms of ASD, the JASPER, and ImPact models stand out for the number of studies. In case of the programs aimed at promoting positive parenting and family welfare, the PCIT model was found in five publications, although evidence of its efficacy is low. Finally, in the group of programs centered on play with children, the FTP model stands out for the number of publications, although the level of evidence of its effectiveness is also low. Based on this, we conclude that there is a need to continue increasing the number of controlled clinical trials with the purpose of reaching the highest level of evidence possible. In any case, the evidence points toward the inclusion of parents in order to gain generalization, thereby increasing the effectiveness of the intervention programs. In a complementary way, programs for the improvement of parent–child interaction are also efficient. 

In general terms, our research allowed us to build on previous results, where it is considered that parental involvement in the therapeutic process can be of great help in increasing social and communication skills in children diagnosed with ASD. As evidenced by the results of the search, there are many different methods and approaches, from training in comprehensive programs to specific programs that impact on parental training from intervention models for the improvement of general parenting. Special attention should be given to programs aimed at improving interaction between parents and children using play as a mediation.

Some of the research analyzed showed that parents increase their knowledge in the face of a diagnosis, improve their parental skills, and generate positive relationships with their children. However, many methods leave aside the support that parents should receive, since they are programs that focus on psychoeducation and direct intervention with children, without taking into account that the vast majority of caregivers present feelings of anxiety and depression, which can hinder the relationship with their children and, as a consequence, the adherence to an intervention program and the achievement of good therapeutic results. In view of this, we consider that to achieve adequate effects, there must be an approach of diverse methodologies that allows the intervention to be implemented with family in an integral manner.

Among the programs we analyzed, there were great differences. Many of them were university clinical trials that must yet be developed a long way before they can be considered evidence. Among other deficiencies, we found little generalization of their use, as many of these clinical trials were developed by the same research team. Additionally, many research centers and universities developed services for the public by developing interventions under one of the described intervention models. These interventions had a double effect, the first of which was improvements in the deficits of children or family relationships, but they also aimed to collect data for future review and analysis. To generalize their use, they provided training for other professionals to use them. Although great efforts have been made to transfer and implement evidence-based intervention strategies to actual community intervention settings [170,171,172], there are difficulties in translating evidence-based practices from university settings into community experiences for various reasons, such as a lack of appropriately qualified technical staff, inadequate settings, or a lack of funding [173,174].

As a general conclusion, we must include parents in interventions with their children, providing them with training and defense strategies to help combat the possible stress associated with the intervention, without forgetting that the main agent is the affected child and, therefore, must be the focus of an intervention. The training of parents should be carried out using all possible resources, e.g., reading of self-administered manuals, meeting and mutual support groups, video feedback sessions, and remote support through the web or video conferences [175]. Without forgetting that the perception of social support derived from the training programs for parents and the interaction and support between them has a great therapeutic value [176]. Comprehensive programs with training for parents, such as the P-ESDM, in which emphasis is placed on the development of play as a therapeutic element, meet the optimum conditions.

## Figures and Tables

**Figure 1 children-07-00294-f001:**
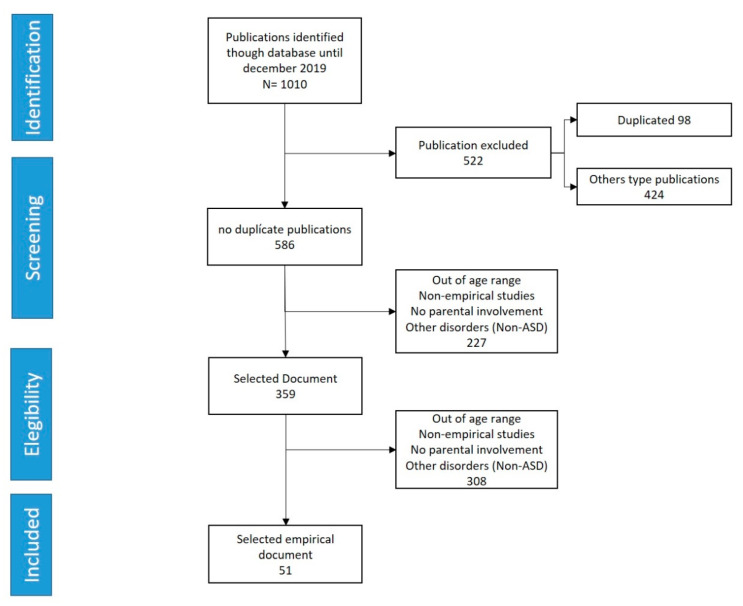
Flow of the document selection process following the inclusion–exclusion criteria. ASD, autism spectrum disorder.

**Figure 2 children-07-00294-f002:**
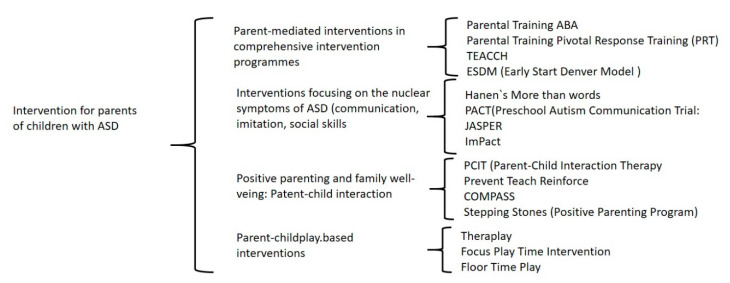
Classification of the 15 programs found in the search.

**Table 1 children-07-00294-t001:** Modified levels of evidence from the original *Journal of Clinical Child and Adolescent Psychology* (JCCAP) criteria.

LEVEL 1	1. Well-established evidence	1. Meta-analysis of randomized clinical trials	Individual data analysis; homogeneous studies; different analysis techniques; meta-regression; meta-analysis; quality of studies
	2. Adequate	2. Randomized clinical trials developed by independent teams	Evaluation of statistical power; multilevel; quality of the studies
LEVEL 2	1. Probably effective	1. Randomized clinical trials with small samples	Evaluation of statistical power; matching controls in time; quality of the studies
	2. Possibly efficient	2. Meta-analysis of single-subject studies with satisfactory results	Individual data analysis; homogeneous studies; different analysis techniques; quality of the studies
LEVEL 3	1. Quasi-experimental	1. Studies of two non-randomized groups with statistically significant results	Quality of the studies
		2. Studies of a single non-randomized group with pre- and post-tests	
		3. Single case studies	
LEVEL 4	1. Questionable effectiveness	1. Qualitative descriptive case studies; uncontrolled clinical series; expert committees	

**Table 2 children-07-00294-t002:** Assessment of the study quality criteria from a methodological point of view.

	Study Quality Criteria	Evaluation
1	Design and assignment of participants to groups: the study includes two group designs with random assignment of participants to the control and treatment groups. The randomization procedure should be specified.	0. No contribution 1. Inadequate 2. Doubtful 3. Adequate
2	Independent variable is well defined: it is properly defined, and manuals or treatment scripts are used.	0. No contribution 1. Inadequate 2. Doubtful 3. Adequate
3	Well-clarified reference population: the study is conducted on a well-defined population, and addresses a specific problem for which the inclusion criteria have been clearly defined.	0. No contribution 1. Inadequate 2. Doubtful 3. Adequate
4	Outcome evaluation: evaluation is done using reliable standardized tests designed to measure the specific problems targeted by the intervention.	0. No contribution 1. Inadequate 2. Doubtful 3. Adequate
5	Adequacy of the statistical analysis: appropriate analysis methods are used, and the sample size is sufficient to detect the studied effects.	0. No contribution 1. Inadequate 2. Doubtful 3. Adequate

**Table 3 children-07-00294-t003:** Search results from 2010 to 2020.

		Subjects		Level of Evidence	Quality of the Study				
	Country **	Sample	Age		1	2	3	4	5
Parent-Mediated Intervention in Comprehensive Intervention Programs									
Search Results for “Parent Training”									
Bearss et al. (2015) [28]	US	180	36 to 84	2.1	2	3	3	2	3
Oosterling et al. (2010) [36]	NL	67	12 to 42	2.1	3	2	3	3	3
Bearss et al. (2013) [37]	US	16	36 to 72	3.2	0	3	3	3	2
Bagaiolo et al. (2017) [38]	BR	67	48	2.1	2	1	2	1	2
Wacker et al. (2013) [39]	US	17	24 to 84	4.2	0	2	3	3	2
Blackman et al. (2020) [40]	US	80	36 to 60	2.1	2	3	3	2	3
Iadarola et al. (2018) [41]	US	108	36 to 84	2.1	3	3	3	3	3
Search Results for “Pivotal Response Training”									
Minjarez et al. (2011) [42]	US	28	24 to 84	2.1	1	2	3	2	2
Gengoux et al. (2015) [43]	US	23	24 to 72	2.1	2	2	2	2	3
Hardan et al. (2015) [44]	US	53	24 to72	2.1	2	2	3	2	3
Bradshaw et al. (2017) [45]	US	3	15 to 21	3.3	0	3	2	2	2
Search Results Classified as TEACCH *									
Welterlin et al. (2012) [46]	US	20	24 to 36	2.1	3	2	2	2	3
D’Elia et al. (2014) [47]	IT	30	24 to 72	2.1	2	2	3	2	3
Search Results Classified as ESDM * Methodology									
Rogers et al. (2012) [48]	US	98	12 to 24	2.1	2	3	2	3	3
Rogers et al. (2014) [49]	US	7	7 to 15	2.1	2	3	3	3	3
Vivanti et al. (2014) [50]	AU	7	24 to 72	2.1	2	2	3	3	3
Rogers et al. (2018) [51]	US	45	12 to 30	2.1	3	2	3	3	3
Waddington et al. (2019) [52]	NZ	5	23 to 59	3.2	0	2	2	3	2
Parent-Mediated Programs Focused on ASD Symptoms									
Search results classified as Hanen program									
Carter et al. (2011) [53]	US	62	20	2.1	2	2	2	3	2
Search Results Classified as the “PACT” Method									
Green et al. (2010) [54]	UK	152	24 to 48	2.1	3	3	3	3	3
Search Results for the JASPER * Model									
Goods et al. (2013) [55]	US	15	36 to 60	2.1	3	3	3	3	3
Kasari et al. (2015) [56]	US	86	22 to 36	2.1	3	3	3	2	3
Gulsrud et al. (2016) [57]	US	86	36	2.1	2	3	3	3	3
Shiere et al. (2016) [58]	US	85	36	2.1	2	3	3	3	3
Search Results for the ImPact * Model									
Ingersoll and Wainer (2013) [59]	US	27	36	2.1	2	3	2	2	3
Ingersoll and Wainer (2013) [60]	US	8	36 to 72	3.3	0	0	2	2	3
Stadnick et al. (2015) [61]	US	16	47	2.1	2	3	3	3	3
Ingersoll et al. (2016) [62]	US	28	19 to 73	2.1	2	3	3	3	3
Ingersoll et al. (2017) [63]	US	9	32 to 65	2.2	0	3	3	3	3
Programs for the Promotion of Positive Parenthood and Family well-being									
Search results for the PCIT * model									
Lesack et al. (2014) [64]	US	1	60	3.3	0	2	2	2	2
Hansen and Shillingsburg (2016) [65]	US	2	45 and 32	3.3	0	2	3	2	2
Masse et al. (2016) [66]	US	3	24 to 84	3.3	0	2	3	3	3
Zlomke et al. (2017) [67]	US	17	24 to 96	3.3	0	3	3	2	2
Parladé et al. (2020) [68]	US	36	36 to 84	2.1	3	3	3	3	3
Search Results for the PRT-F * Model									
Sears et al. (2013) [69]	US	2	36 to 60	3.3	0	1	2	3	3
Bailey et al. (2015) [70]	US	3	60 to 84	3.3	0	2	3	2	3
Search Results for the COMPASS * Method									
Ruble et al. (2011) [71]	US	35	72	3.1	1	3	3	3	2
Kuravackel et al. (2018) [72]	US	33	96	2.1	2	3	3	3	3
Search Results for the SSTP * Method									
Roux et al. (2013) [73]	AU	52	24 to 108	2.1	3	3	2	3	3
Tellegen et al. (2014) [74]	AU	64	24 to 108	2.1	2	3	2	3	3
Schortt et al. (2018) [75]	DE	24	36 to 144	3.1	0	2	3	2	3
Play-Focused Intervention Programs									
Search Results for the Theraplay Method									
Howard et al. (2018) [76]	US	8	36 to 108	3.3	0	2	2	2	2
Search Results for the Floor Time Play Method									
Dionne et al. (2011) [77]	CA	1	42	3.3	0	1	0	1	0
Pajareya and Nopmaneejumruslers (2011) [78] Pajareya and Nopmaneejumruslers (2012) [79]	TH	32	24 to 72	2.1	3	2	2	2	2
Lil and Chhanbria (2013) [80]	TW	11	42 to 69	3.1	0	3	2	2	3
Solomon et al. (2014) [81]	US	128	32 to 72	2.1	3	2	2	2	3
Aali et al. (2015) [82]	IR	12	36 to 96	3.1	0	0	0	1	1
Sealy and Glovinsky (2016) [83]	US	40	24 to 72	2.1	3	2	3	2	3
Search Results for the FPI *									
Siller et al. (2013) [84] Siller et al. (2014) [85]	US	70	36 to 72	2.1	3	2	3	3	2

* TEACCH, Treatment and Education of Autistic and Related Communication-Handicapped Children; ESDM, Early Start Denver Model; PACT, Preschool Autism Communication Trial; JASPER, Joint Attention Symbolic Play, Engagement, and Regulation; ImPact, Improving Parents as Communication Teachers; PCIT, Parent–Child Interaction Therapy; PRT-F, Prevent–Teach–Reinforce for Families; COMPASS, Collaborative Model for Promoting Competence and Success (COMPASS); FPI, Focus Playtime Intervention. ** The international acronyms have been used in the names of the countries.

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
