# Peer review of "Early Intervention with Parents of Children with Autism Spectrum Disorders: A Review of Programs"

_children, 2020, doi:10.3390/children7120294_

Round 1

Reviewer 1 Report

In this systemic review study by Torres et al, authors have analyzed the currently available programs which involve parental involvement towards early intervention with children having ASD. Since, ASD affects a significant number of children, and yet have no definitive cure, its early diagnosis and parental training for these children at their early stage of development are crucial factor to help these children keeping up with mainstream life-style as the grow up. While the present study has provided careful analyses of the important psychological therapy techniques based on the available published scientific evidences, making this article insightful and significant contribution to the field, however, there are certain concerns which need to be addressed to improve this article. Please find the suggestions below;

1) The overall manuscript requires significant English language editing with clearly understandable sentences, e.g. the Introduction part is very difficult to understand, particularly from Line 31 to 56. Please improve the section.

2) In Line 49, please correct the author name for Reference 25.

3) In Line 205, please correct the spelling errors.

4) In Line 574, please format the DIR abbreviation and citations. Citations should be outside parenthesis for full form for DIR.

5) Given the fact the parents' education, socio-economic status and psychological strength are critical factors toward greater success for parental involvement-mediated early intervention program, authors should include these factors in their analyses with inclusion of sufficient and relevant background and proper citations.

Author Response

The manuscript has been reviewed in its entirety by the MDPI for English editing. Typographical and spelling errors have been corrected. We assume reviewer 1's criticism regarding the importance of socio-economic factors and the psychological strength of parents. In this sense, most of the programs we analyzed are precisely aimed at generating psychological well-being to those families that present symptoms of anxiety and stress. The conclusions have been remodeled.

Reviewer 2 Report

Thank you for inviting me to review this paper entitled “Early Intervention with Parents of Children with Autism Spectrum Disorders: A Review of Programs”. This is an interesting systematic review of the type and effectiveness of early intervention programs for parents with children with ASD. Although I acknowledge the efforts of the authors and the importance of the paper, I have some suggestions that could help improve the manuscript. Particularly, I suggest consulting the usual methodology of systematic reviews. More specific comments are reported below.

  • Line 49 “Kim, Bal y Lord [25]”: Spanish typo
  • Inclusion criteria: please, specify inclusion criteria according to the PICOS criteria: Participants, Intervention, Comparison/control, Outcome, Study design.
  • Exclusion criteria: please specify only the criteria and not the selection process (e.g. number of articles excluded with reasons). The selection process must be explained later, at the beginning of Results.
  • Also, the paragraphs about inclusion and exclusion criteria can be merged.
  • Figure 1: Please report reasons for exclusion in the figure.
  • Results should start reporting the study selection process.
  • Table 3: I suggest inserting the first author and year in the first columns, otherwise it is really unclear and difficult to understand which articles the characteristics are referred to.
  • Please, provide a table legend with abbreviations in full.
  • It is unclear if the N participants referred to the entire samples, or only to the parents who were randomized/assigned to the intervention arm.
  • It is also unclear whether the authors have included studies with a control group. Please, specify it in the Inclusion criteria as suggested, and in case of different study design included, also in the table.
  • Apart from the quality of the studies, it would be nice and clearer to have a table including at least the study characteristics, the outcomes and the results. So, the reader doesn’t have to read all the text of results but can just pass through the studies of interest.
  • Discussion. What about the outcomes? Which outcomes were considered in the included studies, and where improvements have been reported? A recent systematic review of 406 clinical trials (doi: 10.1177/1362361319854641) evidenced that outcomes are very sparse in treatments directed to people with ASD. What about parent-mediated treatments? Please, consider the aforementioned paper while discussing this issue.

Author Response

The manuscript has been reviewed in its entirety by MDPI for English editing. The manuscript has been reviewed in its entirety by MDPI for English editing. Typographical and spelling errors have been corrected. Regarding the comments of the reviewer 2, the errors detected have been corrected and the exclusion criteria have been clarified. Table 3 presents an assessment of the quality of the studies related to the design of the trial, participants, data analysis according to the criteria established in Table 1 and 2. In this section (quality assessment) the JCCAP criteria described in Table 1 and 2 are used. Incorporating a new table with the information requested would be, in our opinion, redundant. Table 3 has been improved by incorporating the name of the first author and the year of each paper. The numbering of the citations has been maintained to give coherence to the article. Those studies selected in the search have been marked with an * in the references.
Regarding the conclusions, this is a descriptive study. It is difficult to compare the results from one program to another because the objectives of these programs are different, in some cases it is to improve the symptoms of ASD with direct intervention and in others it is intended to improve the quality of life of families to benefit children with ASD.

Round 2

Reviewer 1 Report

Thanks to the authors for making great efforts toward significant improvement of the manuscript. The manuscript is now suitable for publication in this journal.

Reviewer 2 Report

The authors have addressed my comments and the manuscript is now improved.